# An Exceptional Cause of Increased ^18^F-Fluorodeoxyglucose Uptake on PET/CT in a Thyroid Nodule

**DOI:** 10.3390/diagnostics13020296

**Published:** 2023-01-12

**Authors:** Ringo Manta, Raoul Muteganya, Abraham J. Beun, Jennifer Fallas, Kris G. Poppe

**Affiliations:** 1Department of Nuclear Medicine, CHU Saint Pierre, Université Libre de Bruxelles (ULB), 1000 Brussels, Belgium; 2Department of Internal Medicine, CHU Saint Pierre, Université Libre de Bruxelles (ULB), 1000 Brussels, Belgium; 3Department of Pathology, Institut Jules Bordet, Université Libre de Bruxelles (ULB), 1000 Brussels, Belgium; 4Department of Endocrinology, CHU Saint Pierre, Université Libre de Bruxelles (ULB), 1000 Brussels, Belgium

**Keywords:** thyroid sarcoidosis, FDG PET/CT, thyroid nodule

## Abstract

A 41-year-old female underwent a cervical spine CT for the workup of posterior neck pain irradiating to the shoulders for several months. An incidental thyroid nodule was found and classified as Bethesda III on the Fine-needle aspiration cytology (FNAC) results. Three months later, the patient developed mild shortness of breath, dry cough, and fever. Chest X-ray revealed a mild enlargement in the bilateral hilar regions. CT showed mediastinal and bilateral hilar enlarged lymph nodes and pulmonary micronodules. The workup was further completed with a ^18^F-FDG PET/CT, showing intense FDG uptake in the mediastinal and bilateral hilar lymph nodes and increased uptake in the thyroid nodule. Endobronchial Ultrasound-guided Transbronchial needle aspiration (EBUS-TBNA) of a left hilar lymph node showed epithelioid non-necrotizing granulomas. Because of the FNAC results, size of the nodule and tracheal shift, thyroid lobectomy was performed one month later. Histopathological results also revealed multiple non-necrotizing epithelioid granulomas, suggesting systemic sarcoidosis with involvement of the thyroid. To our knowledge, this is the first report of thyroid sarcoidosis detected on ^18^F-FDG PET/CT. Although an increased FDG uptake in a thyroid nodule is usually suggestive of thyroid malignancy, toxic nodule, or follicular hyperplasia, our case report shows that it could also suggest thyroid sarcoidosis.

**Figure 1 diagnostics-13-00296-f001:**
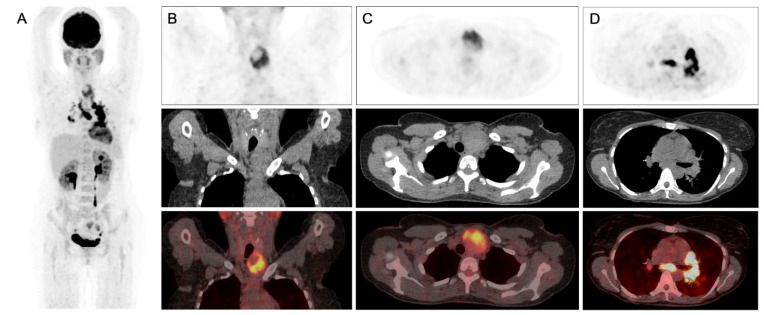
^18^F-FDG PET/CT of the patient: (**A**) Maximum Intensity Projection image, (**B**) coronal images, (**B**) transaxial images of the lower neck (**C**,**D**) and transaxial images of the thorax (PET, upper; CT, middle; fused PET/CT, lower). A 41-year-old female was initially evaluated for posterior neck pain irradiating to the shoulders for several months. She was known to have a heterozygote checkpoint kinase 2 (CHEK2) mutation and had a medical history of high blood pressure and Bell’s palsy in 2017. The family history revealed that her mother was diagnosed with breast cancer at a young age. Computed tomography (CT) of the cervical spine revealed an incidental 56 mm thyroid nodule with a tracheal shift. Thyroid function tests were normal. Fine-needle aspiration cytology showed “Follicular Lesions of Undetermined Significance” (FLUS), also referred to as “Atypia of Undetermined Significance” (AUS) or classified as category III according to the Bethesda system. The patient was set to undergo a thyroid lobectomy, but surgery was delayed due to the COVID-19 pandemic. Three months after the initial presentation, the patient presented with mild shortness of breath, dry cough, and fever for three weeks prior to the visit, but she no longer complained of neck pain. Physical examination showed a palpable nodule of the left thyroid lobe. Laboratory investigations showed a white blood cell count of 4000/mm^3^ (normal range 4000–11,000) and a C-reactive protein (CRP) level of 1.4 mg/L (normal range < 5.0), and the serum angiotensin-converting enzyme level was elevated at 125 U/L (normal range 20–70). Other biological parameters were unremarkable. Polymerase-chain-reaction (PCR) assay and serologic tests showed no evidence of active infection with SARS-CoV-2, influenza A and B. A chest X-ray revealed mild enlargement in the bilateral hilar regions. Chest CT showed bilateral mediastinal and bilateral hilar enlarged lymph nodes and several micronodules in the middle lobe and the left lower lobe. Tuberculin skin test was negative, and the pulmonary function tests were normal. ^18^F-FDG PET/CT shows a heterogeneously increased FDG uptake in the thyroid nodule (SUVmax 9.9) and an intense FDG uptake in mediastinal and bilateral hilar lymph nodes (SUVmax 23.6). A mild FDG uptake in the lower fields of both lungs was also observed. Endobronchial Ultrasound-guided Transbronchial needle aspiration (EBUS-TBNA) of a left hilar lymph node showed epithelioid non-necrotizing granulomas. Sarcoidosis was diagnosed based on the FDG PET/CT findings and the EBUS-TBNA results. Oral methylprednisolone was initiated, soon improving the symptoms. Thyroid lobectomy was performed one month later. Histopathological results of the thyroid specimen also revealed multiple non-necrotizing epithelioid granulomas and no signs of malignancy. These findings along with the thoracic findings on ^18^F-FDG PET/CT suggested sarcoidosis involvement of the thyroid gland. Sarcoidosis is a multisystem granulomatous disease that mainly involves the lungs, thoracic lymph nodes, and the skin. Thyroid gland involvement is rare, accounting for 1 to 4% of patients with systemic sarcoidosis [1,2]. Thyroid sarcoidosis usually presents as diffuse goiter, but multinodular goiter and solitary nodule are also reported [3,4,5]. Symptoms include oppressive thyroid pain, dyspnea and dysphagia [3]. Laboratory results generally show normal thyroid function or hypothyroidism [6]. The final diagnosis of thyroid sarcoidosis is often made with FNAC, after thyroidectomy or autopsy [1,3]. Coexisting thyroid disorders such as Hashimoto’s disease, Grave’s disease and thyroid carcinomas are common [4]. ^18^F-FDG PET/CT has an established role in the evaluation of various malignancies and inflammatory disorders due to an increased glucose transporter expression and upregulation of hexokinase enzymes in neoplastic and inflammatory cells [7]. The clinical use of ^18^F-FDG PET/CT in the diagnosis and management of sarcoidosis is increasing. FDG uptake in sarcoid lesions has been shown to be due to the high glycolytic activity of activated macrophages and lymphocytes [7]. Typical thoracic sarcoidosis findings on ^18^F-FDG PET/CT include intense FDG uptake in mediastinal and bilateral lymph nodes in a symmetrical distribution. ^18^F-FDG PET/CT also allows for the detection of unexpected or clinically silent lesions with high sensitivity. In normal individuals, thyroid FDG uptake is low or absent [8]. Focal thyroid FDG uptake is found in 0.2 to 10.1% of individuals [9]. Approximately one third of high focal FDG in the thyroid is associated with malignancy [10], most of which is papillary carcinoma [8]. Other malignant lesions commonly reported include follicular neoplasm, metastases, medullary thyroid carcinoma, and lymphomas [9,11]. FDG avid thyroid malignancies tend to have higher glycolytic rates and are associated with a more aggressive behavior and a poorer prognosis [8]. Benign lesions with focal thyroid uptake are mostly associated with follicular adenoma and nodular hyperplasia. Other benign thyroid lesions, such as toxic adenoma, Hurtle cell adenoma, or hemorrhagic cyst, are also reported [12,13]. However, a wide overlap in FDG uptake between benign and malignant thyroid lesions is present [10]. To our knowledge, there are no previous reports of thyroid sarcoidosis presenting with an increased thyroid uptake on ^18^F-FDG PET/CT. This case highlights that an increased thyroid uptake on ^18^F-FDG PET/CT in a patient with systemic sarcoidosis might suggest the presence of sarcoidosis involvement of the thyroid gland. However, thyroid malignancy must still be ruled out considering its high prevalence in cases of focal FDG uptake. Whether thyroid sarcoid lesions may regress under systemic corticoid therapy remains unknown and would require further studies.

**Figure 2 diagnostics-13-00296-f002:**
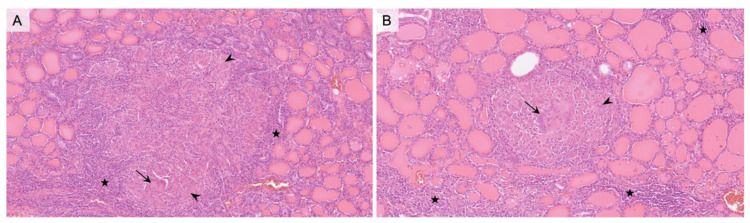
(**A**) 100× and (**B**) 200× hematoxylin- and eosin-stained sections of the thyroid lobectomy showing multiple scattered non-necrotizing epithelioid granulomas located in the interstitial tissue, with Langhans giant cells (*arrows*), epithelioid histiocytes (*stars*), and peripheral lymphocytic infiltrate (*arrowheads*). Our patient’s histopathological result from the thyroid specimen showed no signs of neoplasia. Non-necrotizing epithelioid granulomas were identified both in the thyroid specimen and in the left hilar lymph node, suggesting systemic sarcoidosis with involvement of the thyroid gland. The presence of non-necrotizing epithelioid granulomas is the pathologic hallmark feature of sarcoidosis. However, non-necrotizing granulomas are not specific for sarcoidosis and should be differentiated from other causes of granulomas such as tuberculosis and De Quervain’s Thyroiditis, but the latter was not suspected based on the clinical and biological data. Additionally, granulomas in De Quervain’s thyroiditis are typically found around the follicles, and giant cells surround a core of colloid, unlike sarcoid granulomas [6]. Tuberculosis was otherwise ruled out based on the lack of risk of exposure to the disease and the negative tuberculin test.

## Data Availability

Not applicable.

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
