# Peer review of "An Exceptional Cause of Increased 18F-Fluorodeoxyglucose Uptake on PET/CT in a Thyroid Nodule"

_diagnostics, 2023, doi:10.3390/diagnostics13020296_

Round 1
Reviewer 1 Report
I find the diagnostic images you would like to publish interesting. I find the diagnostic pictures you would like to publish interesting. As my general suggestion, the authors should put what is written in the captions, i.e. the description of the clinical history, in a separate paragraph. In the caption it should be better explained what is observed from the imaging. Also, in some passages the information provided is not clear. It is requested to be more precise and comprehensive. It is advisable to include figure 1 and figure 2 in the text, as is done for scientific manuscripts, so that it is clear what time frame the images refer to. Finally, in the abstract the authors mention histopathological findings indicative of a sarcoi... but then in the body of the article the histopathological examination is implied by showing only figure 2. Again, it should be written/described in the text what is observed in the histopathological samples shown in figure 2
In detail:
line 43 The authors write: three months later..... I wonder with respect to which time? To the date on which the operation was scheduled? At the end of the first wave of sars cov2? It is unclear
Line 45: The authors report the following phrase: Physical examination revealed a palpable nodule of the left thyroid lobe. I ask: But was it not diagnosed earlier and precisely because of the presence of this nodule, surgical removal of the thyroid was planned, later postponed because of the pandemic?
I should add, from figure 2 the removal was performed but it is not clear when... after the 18F-FDG PET/CT?
Line 58: The authors should explain in more detail on the basis of which clinical data sarcoidosis was diagnosed. On the basis of EBUS-TBNA?
Other minor editing corrections.
In the caption of figure 1 write the letters in capital letters and put them first as done in figure 2. Add arrows to indicate structures with high uptake of interest
Place references in square brackets
Add space between words and brackets in lines 61 and 64
Author Response
# Line 43 : The authors write: three months later..... I wonder with respect to which time? To the date on which the operation was scheduled? At the end of the first wave of sars cov2? It is unclear
Answer : "Three months after the initial presentation" (which was neck pain) [line 55 of the revised document]
This has been changed in the manuscript.
Line 45: The authors report the following phrase: Physical examination revealed a palpable nodule of the left thyroid lobe. I ask: But was it not diagnosed earlier and precisely because of the presence of this nodule, surgical removal of the thyroid was planned, later postponed because of the pandemic?
I should add, from figure 2 the removal was performed but it is not clear when... after the 18F-FDG PET/CT?
Answer : "Physical examination showed" The term "revealed" was wrongly chosen. This has been changed. [line 58 of the revised document]
# Line 58: The authors should explain in more detail on the basis of which clinical data sarcoidosis was diagnosed. On the basis of EBUS-TBNA?
Sarcoidosis was diagnosed based on the FDG PET/CT findings along with the EBUS-TBNA results. Oral methylprednisolone was initiated, soon improving the symptoms. Thyroid lobectomy was performed one month later.
This has been added in the manuscript. [lines 71-73 of the revised document].
A few words have also been added to clarify and to conclude on that matter [lines 103-108 of the revised document]
Thank you for these crucial remarks.
Sincerely,
Dr Ringo Manta
Corresponding Author
Reviewer 2 Report
Interesting case report and images. I only have a few comments.
Abstract line 15-16 it should be mentioned that the Bethesda grading is based on fine needle aspiration. It know this may be due to word limitations, but in the current sentence it sounds like Bethesda classification was based on CT findings.
Abstract Line 23 - FNAC should be spelled out
Figure 1 legend:
In line 36 and 56 "is" should be replaced with "was" to keep the past tense consistently throughout the manuscript.
I miss some information regarding sarcoidosis involvement of the thyroid gland - how often is that found? Can it cause thyroid dysfunction?
If allowed, more references regarding the frequency of FDG-avid thyroid nodules and the risk of malignancy could be added.
Author Response
Comment #1 Abstract line 15-16 it should be mentioned that the Bethesda grading is based on fine needle aspiration. It know this may be due to word limitations, but in the current sentence it sounds like Bethesda classification was based on CT findings.
Answer : Absolutely. This has been changed in the manuscript [line 29 of the revised document]
# Comment 2 Abstract Line 23 - FNAC should be spelled out
Answer : "Fine-needle aspiration cytology" has been spelled out earlier already following to your first comment's modification. Thank you.
# Comment 3 In line 36 and 56 "is" should be replaced with "was" to keep the past tense consistently throughout the manuscript.
Answer : This has been changed in the manuscript.
# Comment 4 I miss some information regarding sarcoidosis involvement of the thyroid gland - how often is that found? Can it cause thyroid dysfunction?
Answer : Indeed. The manuscript was at first supposed to be in the format of a case report but the suggested format was in the end "Interesting Images", leading us to remove some introduction elements.
We added this information with great pleasure to the manuscript [line 77-84 of the revised document + references].
Thank you for your interest and consideration for our manuscript.
Sincerely,
Dr Ringo Manta
Corresponding Author
Reviewer 3 Report
This case report showed that sarcoidosis invaded the left lobe of the thyroid gland depicted by FDG-PET/CT.
My concerns in this report.
How could the author prove that sarcoidosis invaded (or arose in) the thyroid gland, instead of just surrounding the thyroid gland?
In PET/CT imaging, round FDG absent area can be seen. In the CT image, CT density is increased in this area compared to the surrounding where FDG uptake is confirmed. Generally, a area with increased CT density can be regarded as normal thyroid grand.
The author should present both macroscopic and microscopic pathological images to prove it.
The author did not show any epidemiological information regarding the sarcoidosis involved in thyroid grand, The author stated that “This is the first report of thyroid sarcoidosis detected on 18F-FDG PET/CT”, so the readers should understand the frequency of this in a clinical situation to be able to decide the priority of differential diagnosis in the PET/CT diagnostic reports.
English language edit should be done.
Author Response
Comment #1 : How could the author prove that sarcoidosis invaded (or arose in) the thyroid gland, instead of just surrounding the thyroid gland? In PET/CT imaging, round FDG absent area can be seen. In the CT image, CT density is increased in this area compared to the surrounding where FDG uptake is confirmed. Generally, a area with increased CT density can be regarded as normal thyroid grand.The author should present both macroscopic and microscopic pathological images to prove it.
Answer : Thank you for this insightful remark. The answer is, as the reviewer said, to confront macroscopic and microscopic images. In the histopathology report, several nodules containing numerous granulomas are observed. As found in the literature (and this has been added in the manuscript following comment #2, [line 77-84 of the revised document + references], thank you for that) thyroid sarcoidosis usually presents as diffuse goiter, but multinodular goiter and solitary nodule are also reported. Based on the histopathology reports and the FDG PET/CT findings, we believe that the large nodule that was found is filled with sarcoid granulomas. Regarding the FDG absent area, it might be a normal gland as you said, this remains unclear.
Comment #2 : The author did not show any epidemiological information regarding the sarcoidosis involved in thyroid grand, The author stated that “This is the first report of thyroid sarcoidosis detected on 18F-FDG PET/CT”, so the readers should understand the frequency of this in a clinical situation to be able to decide the priority of differential diagnosis in the PET/CT diagnostic reports.
Answer : Epidemiological information has been included , for a better understanding of the reader [lines 77-84 of the revised document+ references].
Thank you for this remark.
Comment #3 : English language edit should be done
Answer : This will be adressed for revision. Thank you.
Lastly, I would like to thank you for your time and consideration while reviewing our manuscript. This will definitely help us clarify the content.
Sincerely,
Dr Ringo Manta
Corresponding Author
Round 2
Reviewer 1 Report
I thank the authors for accepting my comments and making the corresponding corrections. The work has improved and can be published in the following form.
Reviewer 3 Report
No further comments.